# Structural and biochemical analyses of the DEAD-box ATPase Sub2 in association with THO or Yra1

Yi Ren*, Philip Schmiege, Günter Blobel*

Laboratory of Cell Biology, Howard Hughes Medical Institute, The Rockefeller University, New York, United States

**Abstract** mRNA is cotranscrptionally processed and packaged into messenger ribonucleoprotein particles (mRNPs) in the nucleus. Prior to export through the nuclear pore, mRNPs undergo several obligatory remodeling reactions. In yeast, one of these reactions involves loading of the mRNA-binding protein Yra1 by the DEAD-box ATPase Sub2 as assisted by the hetero-pentameric THO complex. To obtain molecular insights into reaction mechanisms, we determined crystal structures of two relevant complexes: a THO hetero-pentamer bound to Sub2 at 6.0 Å resolution; and Sub2 associated with an ATP analogue, RNA, and a C-terminal fragment of Yra1 (Yra1-C) at 2.6 Å resolution. We found that the 25 nm long THO clamps Sub2 in a half-open configuration; in contrast, when bound to the ATP analogue, RNA and Yra1-C, Sub2 assumes a closed conformation. Both THO and Yra1-C stimulated Sub2's intrinsic ATPase activity. We propose that THO surveys common landmarks in each nuclear mRNP to localize Sub2 for targeted loading of Yra1.

*For correspondence: yren@ rockefeller.edu (YR); blobel@ rockefeller.edu (GB)

**Competing interests:** The authors declare that no competing interests exist.

## Introduction

Eukaryotic gene expression requires a series of highly coordinated nuclear events that include transcription, pre-mRNA processing, and packaging of mRNPs for nuclear export. The evolutionarily conserved TREX (TRanscription-EXport) complex, which facilitates assembly of export competent mRNPs and functionally connects transcription with nuclear export, is a key player in these events (*Masuda et al., 2005*; *Rehwinkel et al., 2004*; *Rodríguez-Navarro and Hurt, 2011*; *Strässer et al., 2002*).

The TREX complex in yeast consists of the hetero-pentameric THO complex, the DEAD-box ATPase Sub2 and the mRNA-binding protein Yra1 (*Chávez et al., 2000*; *Strässer and Hurt, 2000, 2001*). TREX travels along actively transcribed genes with RNA polymerase II (Pol II) (*Meinel et al., 2013*; *Strässer et al., 2002*). Mature nuclear mRNPs, which have undergone capping, splicing and polyadenylation, are potential 'substrates' for TREX. In essence, TREX-mediated remodeling yields a targeted deposition of Yra1 on nuclear mRNP (*Abruzzi et al., 2004*), which in turn serves as a platform to bind hetero-dimeric Mex67•Mtr2 (*Strässer and Hurt, 2000*; *Zenklusen et al., 2001*). The latter, being capable of binding to Phe-Gly (FG) repeat regions of nucleoporins, provides access to the nuclear pore complex (NPC) (*Fribourg et al., 2001*; *Segref et al., 1997*). Although Mex67•Mtr2 is not a bona fide member of the TREX complex, Mex67 binds via its ubiquitin-associated (UBA) module to the Hpr1 subunit of THO (*Gwizdek et al., 2006*). Collectively, these data suggest that TREX-mediated reactions yield a targeted and concerted deposition of both Yra1 and Mex67•Mtr2 on nuclear mRNP in preparation for its export through the NPC.

The Sub2 ATPase, belonging to the DEAD-box family of RNA-protein complex remodeling enzymes, is central for these TREX-mediated reactions. Previous studies showed that Sub2

association with active genes requires an intact THO complex, and it is thought to be transferred to the mRNA during cotranscriptional mRNP packaging (*Abruzzi et al., 2004*; *Zenklusen et al., 2002*). In addition, Sub2 and Mex67•Mtr2 bind to the same regions of Yra1, suggesting that Sub2 could recruit Yra1 onto mRNP and then be displaced by Mex67•Mtr2 (*Strässer and Hurt, 2001*). These observations raise the question as to how TREX coordinates Sub2 ATP hydrolysis with the deposition of Yra1 and Mex67•Mtr2 on mRNP.

Here, we report crystal structures of two relevant assemblies to provide insights into the mRNP-remodeling reactions carried out by TREX. First, a 6.0 Å resolution structure of a 360 kDa THO complex bound to Sub2 revealed that THO clamps the two lobes of Sub2 in a half-open configuration, in the middle of its 25 nm elongated structure. Second, a 2.6 Å structure of Sub2 bound to an ATP analogue, poly (U) RNA, and a C-terminal fragment of Yra1 showed the two lobes of Sub2 in a closed configuration. We suggest that THO serves to survey features of mature nuclear mRNP to mediate an ATP-dependent targeted deposition of Yra1 and Mex67•Mtr2, thereby initiating a series of reactions that are obligatory for export of nuclear mRNP across the NPC.

## Results

### Structural determination of a THO•Sub2 complex

THO is required for the recruitment of Sub2 onto actively transcribed genes (*Zenklusen et al., 2002*). However, the molecular role of their interaction in mRNP remodeling is not clear. To characterize the THO-Sub2 interaction, we set out to assemble the yeast THO complex using an insect cell expression system. Previous studies showed that THO exists as a robust structural and functional unit in vivo, and knockdown of individual subunits causes down-regulation of the other subunits (*Huertas et al., 2006*). Consistent with this, we could only obtain THO by means of co-expression. Because of the low yield of full-length THO, we removed potentially disordered regions based on secondary structure prediction, and replaced the *Saccharomyces cerevesiae* Tex1 subunit with *Saccharomyces bayanus* Tex1, to assemble a stable core module of THO (denoted by THO*). The purified THO* showed stoichiometric amounts of each subunit (*Figure 1A,B*).

THO* was crystallized in complex with separately expressed full-length Sub2. We determined a THO*•Sub2 structure at 6.0 Å resolution using single-wavelength anomalous diffraction (SAD) from crystals soaked with phosphotungstate or tantalum bromide clusters (*Figure 1C*, *Table 1*, and *Figure 1—figure supplement 1*). Although the resolution is not high enough to trace individual residues, it allows us to characterize secondary structures. In this structure, two RecA domains of Sub2, Sub2-N and Sub2-C, are modeled separately using the Sub2 structure reported in this study; a polyalanine model of Tex1 is derived from WDR5, which shares 41% sequence homology with Tex1 for the residue range 50–344; and the remainder of THO is built with polyalanine α–helices. The THO*•Sub2 model presented here reveals the overall architecture of THO and, more importantly, the interaction between THO and Sub2.

### Architecture of THO

The 360 kDa THO*•Sub2 complex reveals an elongated THO*, ~25 nm from 'head' to 'tail' (*Figure 1C*). Its dimensions are close to those reported previously for full-length THO isolated from yeast and analyzed by negative stain EM (*Peña et al., 2012*). Two elements project from the elongated base, an extended one close to the head and a short one that represents the Tex1 subunit. Tex1 folds into a seven-bladed β-propeller with its smaller top surface facing toward the head. It is clamped to the base via contacts on two opposite sides of the propeller.

To further explore the subunit architecture of THO, we mapped the locations of its individual subunits by negative stain EM through N- or C-terminal maltose-binding protein (MBP) labeling (*Figure 1—figure supplement 2*). We found that the N-termini of Tho2 and Hpr1 were localized close to the head, while their C-termini were localized at the tail. These data indicate that the Tho2 and Hpr1 subunits span the longest dimension of THO*. In addition, the N-termini of Mft1 and Thp2 were assigned to the head, while their C-termini could not be positioned reliably.

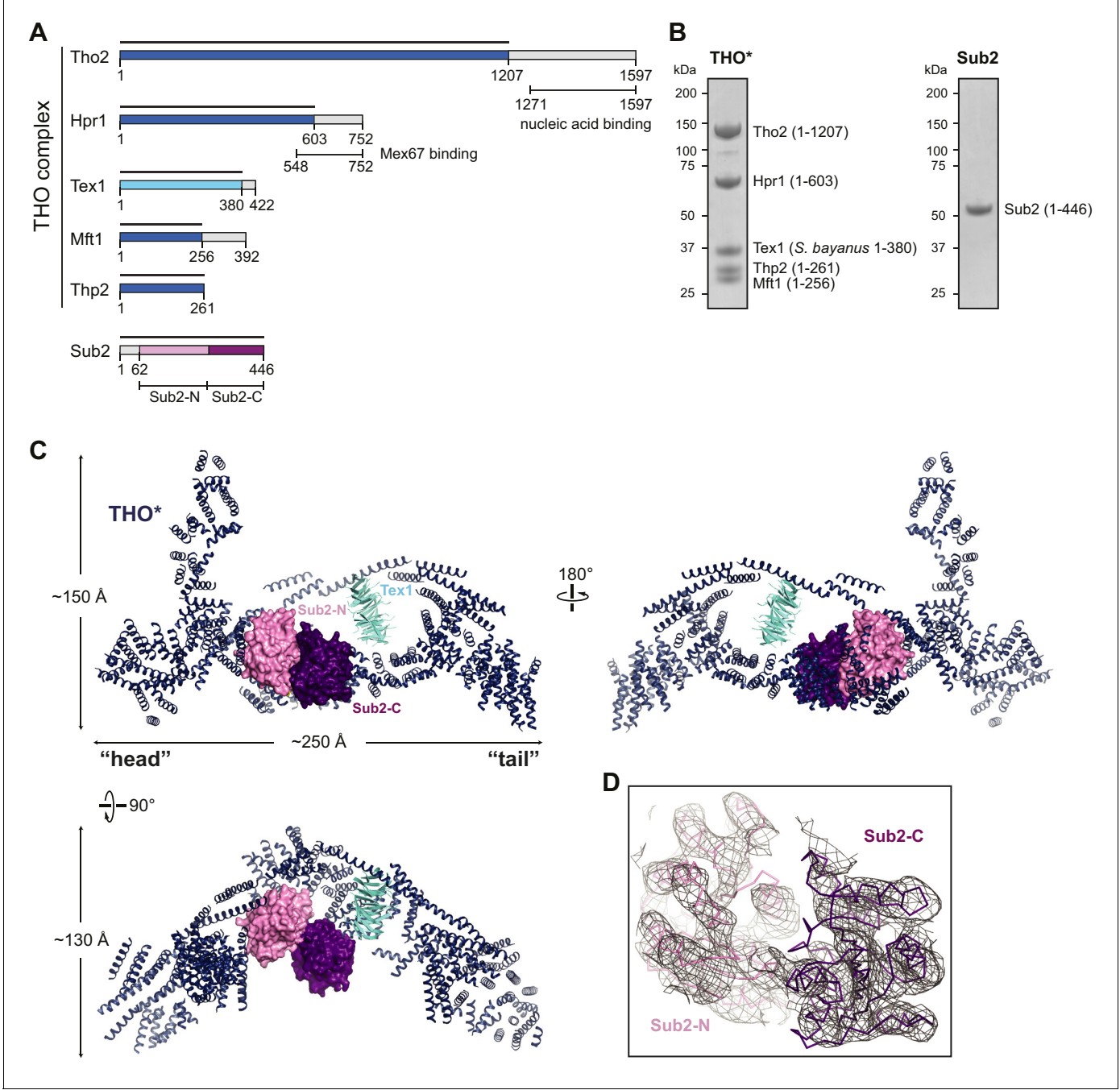

**Figure 1.** Structural overview of the THO*•Sub2 complex. (**A**) Schematic representation of THO and Sub2. Recombinant protein fragments that were used for crystallization are indicated by bars. The THO core complex (THO*) contains *S. cerevisiae* Tho2, Hpr1, Mft1, and Thp2, as well as *S. bayanus* Tex1 (81% sequence identity with *S. cerevisiae* Tex1). The DEAD-box ATPase Sub2 contains two recombinase A (RecA)-like domains labeled as Sub2-N and Sub2-C. (**B**) Coomassie-stained SDS-polyacrylamide gel electrophoresis (PAGE) of purified THO* complex and full-length Sub2. (**C**) Three views of the THO*•Sub2 complex with THO* in a cartoon representation and Sub2 in a surface representation. (**D**) Experimental electron density map of Sub2 after density modification, contoured at 1.0σ level corresponding to the view of the upper left panel in **C**.

The following figure supplements are available for figure 1:

**Figure supplement 1.** Crystallization of the THO*•Sub2 complex.

**Figure supplement 2.** Subunit architecture of THO.

**Table 1.** Data collection and refinement statistics for the structure of the THO*•Sub2 complex.

| | W-I | W-II | W-III | Ta-I | Ta-II |
|---|---|---|---|---|---|
| **Data collection** | | | | | |
| Space group | $P2_1$ | $P2_1$ | $P2_1$ | $P2_1$ | $P2_1$ |
| Cell dimensions | | | | | |
| $a, b, c$ (Å) | 152.9, 323.0, 176.8 | 153.3, 319.5, 176.4 | 152.7, 322.2, 175.6 | 153.3, 327.3, 175.2 | 153.3, 328.1, 174.4 |
| $\beta$ (°) | 101.3 | 101.0 | 101.1 | 101.9 | 101.9 |
| Wavelength (Å) | 1.2123 | 1.2141 | 1.2141 | 1.2524 | 1.2553 |
| Resolution (Å) | 50–6.0 | 50–6.8 | 50–7.2 | 50–7.6 | 50–7.5 |
| | (6.1–6.0)* | (6.92–6.80)* | (7.32–7.20)* | (7.73–7.60)* | (7.63–7.50)* |
| $R_{merge}$ (%) | 14.0 (75.1) | 18.2 (112.7) | 17.7 (87.6) | 11.5 (85.2) | 11.2 (90.5) |
| $<I / \sigma_I>$ | 13.3 (2.1) | 14.4 (2.7) | 11.5 (2.5) | 17.3 (2.2) | 17.6 (2.1) |
| Completeness (%) | 98.8 (94.8) | 89.8 (78.3) | 99.8 (100.0) | 100.0 (100.0) | 100.0 (100.0) |
| $CC_{1/2}$ | 99.3 (68.5) | 99.5 (70.2) | 99.1 (72.8) | 99.7 (77.4) | 99.7 (79.8) |
| Redundancy | 6.5 (4.8) | 9.8 (9.7) | 7.1 (7.2) | 6.8 (6.8) | 6.8 (6.8) |
| **SAD phasing** | | | | | |
| Resolution | 50–8.5 | 50–7.5 | 50–8.0 | 50–8.0 | 50–8.0 |
| Number of sites | 11 | 19 | 19 | 18 | 18 |
| Figure of Merit | 0.41 | 0.45 | 0.46 | 0.45 | 0.45 |
| **Refinement** | | | | | |
| Resolution (Å) | 50–6.0 | | | | |
| No. reflections | | | | | |
| total | 41038 | | | | |
| test set | 2078 | | | | |
| $R_{work}$ / $R_{free}$ (%) | 43.6/43.4 | | | | |
| No. atoms | | | | | |
| Protein | 30205 | | | | |
| Ligand/ion | 583 | | | | |
| B-factors | 200 | | | | |
| Ramachandran plot (%) | | | | | |
| Favored/Allowed/Disallowed | 99.4/0.6/0.0 | | | | |
| R.m.s deviations | | | | | |
| Bond lengths (Å) | 0.010 | | | | |
| Bond angles (°) | 1.408 | | | | |

*Highest-resolution shell is shown in parentheses.

## THO stimulates the ATPase activity of Sub2

The THO*•Sub2 complex features a bi-lobed structure for Sub2 (*Figure 1C* and *Figure 2A*). Interestingly, the middle region of THO* makes contact with both the Sub2-N and Sub2-C lobes and induces a semi-open conformation. DEAD box ATPases like Sub2 exhibit distinct conformations during the ATP hydrolysis cycle. In the absence of nucleotide and RNA, the two RecA domains are separated, representing an 'open' conformation (*Caruthers et al., 2000*; *Shi et al., 2004*). ATP and RNA binding induces a 'closed' form that is productive for ATP hydrolysis (*Sengoku et al., 2006*; *von Moeller et al., 2009*). The semi-open conformation of Sub2 in the THO*•Sub2 complex is close to its ATP/RNA-bound active state (*Figure 2A* and *Figure 2—figure supplement 1*). This

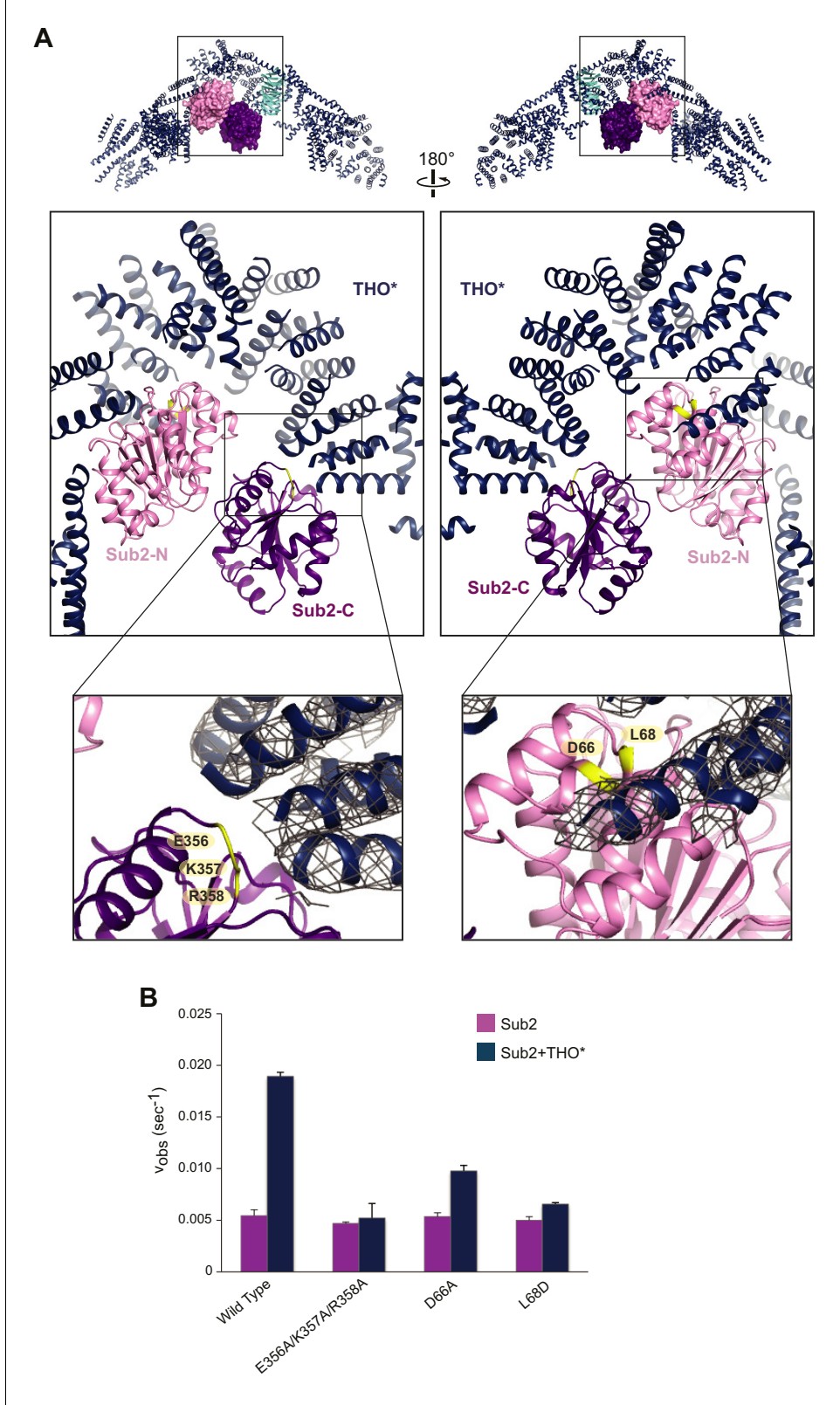

**Figure 2.** Structural basis of Sub2 stimulation by the THO complex. (**A**) THO* binding induces a half-open conformation of Sub2. The upper panels indicate the positions of corresponding middle panels within the THO*•Sub2 complex. The Tex1 subunit of THO* does not contact Sub2 and is omitted in the middle panels for clarity. Interacting residues of Sub2, which have been subjected to mutagenesis in **B**, are highlighted in green. The

*Figure 2 continued on next page*

*Figure 2 continued*

lower panel close-up views show the experimental density map of THO* after density modification, contoured at 1.5σ level. (**B**) Disruption of the THO*-Sub2 interaction compromises the stimulation of Sub2 ATPase activity (mole of ADP produced per second per mole of Sub2) by THO*. Error bars represent standard deviation of three independent experiments.

The following figure supplements are available for figure 2:

**Figure supplement 1.** Different conformations of Sub2 during the ATP hydrolysis cycle.

**Figure supplement 2.** THO and Yra1-C jointly regulate the ATPase activity of Sub2.

**Figure supplement 3.** Structural comparison of THO*•Sub2 to Gle1•Dbp5, eIF4G•eIF4A, and CNOT1•DDX6.

observation prompted us to examine whether THO regulates the Sub2 ATPase activity. Indeed, we found that THO* accelerated the ATPase activity of Sub2 by ~3.5-fold (*Figure 2B*). The Sub2 stimulating activity of THO* is similar to that of full-length THO, indicating that THO* maintains the full stimulatory effect of THO (*Figure 2—figure supplement 2*). Collectively, our results demonstrate that THO is a previously uncharacterized regulator of the Sub2 ATPase.

Notably, this ATPase activation mechanism by configuring the two RecA domains in a half-open conformation is shared with several other DEAD-box proteins: Dbp5 in mRNA export, eIF4A in translation initiation, and DDX6 in miRNA-mediated translational repression, which are activated by Gle1, eIF4G, and CNOT1, respectively (*Figure 2—figure supplement 3*) (*Folkmann et al., 2011*; *Mathys et al., 2014*; *Montpetit et al., 2011*; *Schütz et al., 2008*). Moreover, the THO* arrangement of pairs of helices at the THO*-Sub2 interface resembles the MIF4G (middle of eIF4G) domains from Gle1, eIF4G, and CNOT1, further indicating a conserved mechanism of DEAD-box ATPase activation.

Recognition of the C-terminal RecA domain of DEAD-box proteins by their MIF4G containing activators is the primary interaction critical for activation of ATP hydrolysis (*Hilbert et al., 2011*; *Montpetit et al., 2011*). At the Sub2-C lobe, a loop formed by E356, K357, and R358 faces THO* (*Figure 2A*). We found that the ATPase activity of a Sub2 E356A/K357A/R358A mutant was no longer activated by THO*, indicating the critical role of the interaction between THO and Sub2-C (*Figure 2B*). At the Sub2-N lobe, THO* recognizes a groove formed by two helices of Sub2 (*Figure 2A*). D66A and L68D mutants of Sub2 at this interface compromised the stimulatory effect of THO* (*Figure 2B*). Together, these structure-guided mutagenesis studies demonstrate that the stimulation of Sub2 ATPase activity by THO requires interactions with both lobes of Sub2.

The semi-open configuration of Sub2 bound to THO may represent a 'primed' state for efficient mRNP engagement. DEAD-box proteins like Sub2 lack RNA specificity, so they are directed to their physiological RNA substrates by their partners (*Linder and Jankowsky, 2011*). THO has recently been shown to directly bind the phosphorylated C-terminal domain (CTD) of the largest Pol II subunit (*Meinel et al., 2013*). In addition, cotranscriptional loading of Sub2 onto active genes requires the Hpr1 subunit of THO (*Zenklusen et al., 2002*). Therefore, THO is well suited to the role of recruiting Sub2 to the transcription machinery, priming and targeting it to appropriate mRNP substrates, potentially those that have been properly processed.

## Structural characterization of a Sub2•Yra1-C•RNA complex

Sub2 has been implicated in loading of Yra1 onto mRNP in an ATP-dependent manner from studies on their human homologs (*Dufu et al., 2010*; *Luo et al., 2001*; *Taniguchi and Ohno, 2008*). To understand the molecular basis of Yra1 deposition by Sub2, one of the TREX-mediated remodeling reactions, we sought to determine the structure of their complex. Yra1 contains two conserved motifs at both termini (N-box and C-box), which are involved in Sub2 binding (*Figure 3A*). The two motifs of Yra1 are joined to the central conserved RRM domain through variable regions (N-vr and C-vr), which are implicated in RNA binding (*Hautbergue et al., 2008*; *Zenklusen et al., 2001*). Full-length Yra1 was not soluble under normal buffer conditions. Genetic studies indicated that the N-box and C-box motifs are functionally redundant, but a Yra1 mutant lacking the C-box exhibits a

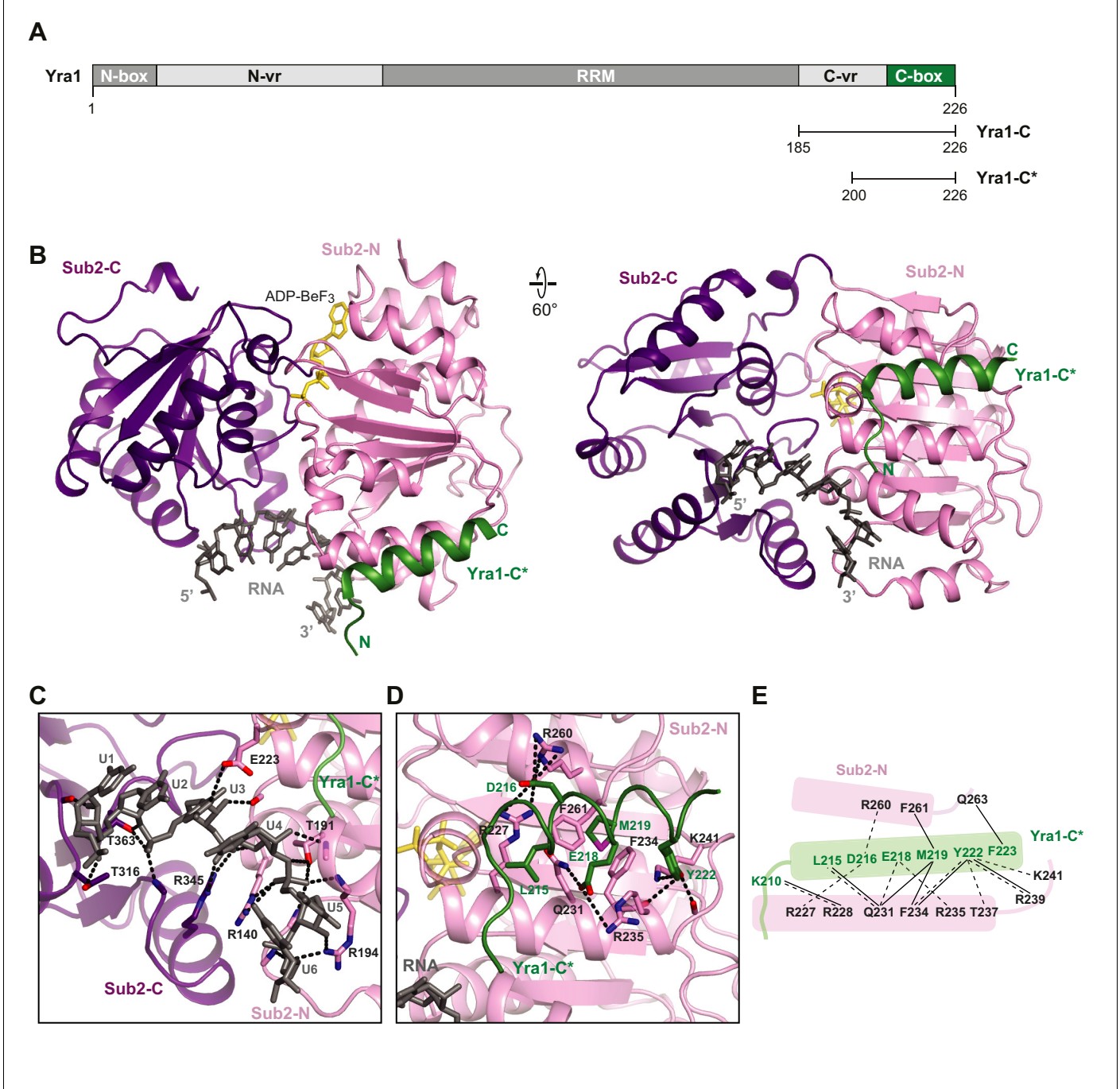

**Figure 3.** Crystal structure of the Sub2•Yra1-C*•RNA complex. (**A**) Schematic representation of Yra1. Yra1-C and Yra1-C* were used for biochemical assays and crystallization studies, respectively. (**B**) Cartoon representation of the Sub2•Yra1-C*•RNA complex in two orientations. (**C** and **D**) Details of the Sub2-RNA and Sub2-Yra1 interactions, corresponding to the view in the right panel in **B**. The polar interaction network is indicated by black dashes. (**E**) Schematic representation of the Sub2-Yra1 interactions. Black dashes indicate polar interactions. Black lines indicate van der Waals interactions.

The following figure supplements are available for figure 3:

**Figure supplement 1.** Comparison of RNA binding by DEAD-box proteins.

**Figure supplement 2.** Multispecies sequence alignment of Yra1-C.

stronger growth phenotype (*Zenklusen et al., 2001*). Thus, we focused on a C-terminal fragment of Yra1, Yra1-C (a.a. 185–226) that comprises C-vr and C-box, to investigate its interaction with Sub2.

We determined a 2.6 Å crystal structure of Sub2 (a.a. 62–446) in complex with a truncated Yra1-C (Yra1-C*, a.a. 200–226), poly (U) 15-mer RNA, and the non-hydrolyzable ATP analogue ADP•BeF$_3$ (*Figure 3B* and *Table 2*). Within this Sub2•Yra1-C*•RNA complex, Sub2 adopts a closed conformation with ADP•BeF3 and 6-nt of RNA (U1 to U6) bound in between Sub2-N and Sub2-C (*Figure 3B, C*, and *Figure 2—figure supplement 1*). The RNA is sharply bent between U4 and U5, as observed in structures of other RNA-bound DEAD-box ATPases (*Figure 3—figure supplement 1*). Sub2 is recognized by the conserved C-box of Yra1 (a.a. 208–226), which folds into a helix that fits in the groove formed by two helices of Sub2-N (*Figure 3—figure supplement 2*). Likely due to flexibility,

**Table 2.** Data collection and refinement statistics for the structure of the Sub2•Yra1*-C•RNA complex.

|  | Sub2•Yra1-C*•RNA |
|---|---|
| **Data collection** | |
| Space group | P3$_1$21 |
| Cell dimensions | |
| *a, b, c* (Å) | 99.3, 99.3, 247.5 |
| *α, β, γ* (°) | 90, 90, 120 |
| Wavelength (Å) | 0.9792 |
| Resolution (Å) | 50–2.6 (2.69–2.60)* |
| $R_{merge}$ (%) | 16.4 (101.6) |
| $<I / \sigma_I>$ | 13.3 (2.3) |
| Completeness (%) | 100.0 (100.0) |
| CC$_{1/2}$ | 99.4 (72.9) |
| Redundancy | 9.9 (8.9) |
| **Refinement** | |
| Resolution (Å) | 50–2.6 |
| No. reflections | |
| total | 44,413 |
| test set | 2235 |
| $R_{work}$ / $R_{free}$ (%) | 21.9/26.8 |
| No. atoms | |
| Protein | 9644 |
| RNA | 360 |
| Ligand/ion | 96 |
| Water | 27 |
| B-factors | |
| Protein | 56.3 |
| RNA | 68.1 |
| Ligand/ion | 46.2 |
| Water | 34.9 |
| Ramachandran plot (%) | |
| Favored/Allowed/Disallowed | 97.6/2.4/0.0 |
| R.m.s deviations | |
| Bond lengths (Å) | 0.007 |
| Bond angles (°) | 1.076 |

*Highest-resolution shell is shown in parentheses.

electron density of the C-vr region and 9-nt of RNA is missing in the crystal structure. The interactions between Sub2 and Yra1 employ a number of Yra1 residues (*Figure 3D,E*). Specifically, Yra1 D216, E218, and Y222 make key polar contacts with Sub2, whereas L215, M219, and Y222 contribute to van der Waals interactions at the interface.

## Sub2 and Yra1 cooperatively bind to RNA

Sub2, like other DEAD-box proteins (*Linder and Jankowsky, 2011*), predominantly interacts with the phosphate-ribose backbone of RNA, with Sub2-N and Sub2-C contacting the 3' and 5' ends, respectively (*Figure 3B,C*, and *Figure 3—figure supplement 1*). Notably, the RNA binding C-vr region of Yra1 that precedes C-box is located close to the RNA, which raises the possibility that it could extend the RNA-binding site in the Sub2•Yra1-C•RNA complex. Therefore, we examined whether Yra1 affects the RNA binding properties of Sub2.

Using an electrophoretic mobility shift assay with poly(U) RNA, we found that Sub2 and RNA formed a complex in the presence of the non-hydrolyzable ATP analogue, ATP-γ-S (*Figure 4A*, compare lanes 1 and 2). In addition, Yra1-C is capable of binding RNA, while C-box alone is not, indicating that the RNA-binding ability indeed lies in C-vr (*Figure 4A*, compare lanes 3 and 8). The presence of Yra1-C promoted Sub2-RNA interaction, forming a trimeric Sub2•Yra1-C•RNA complex

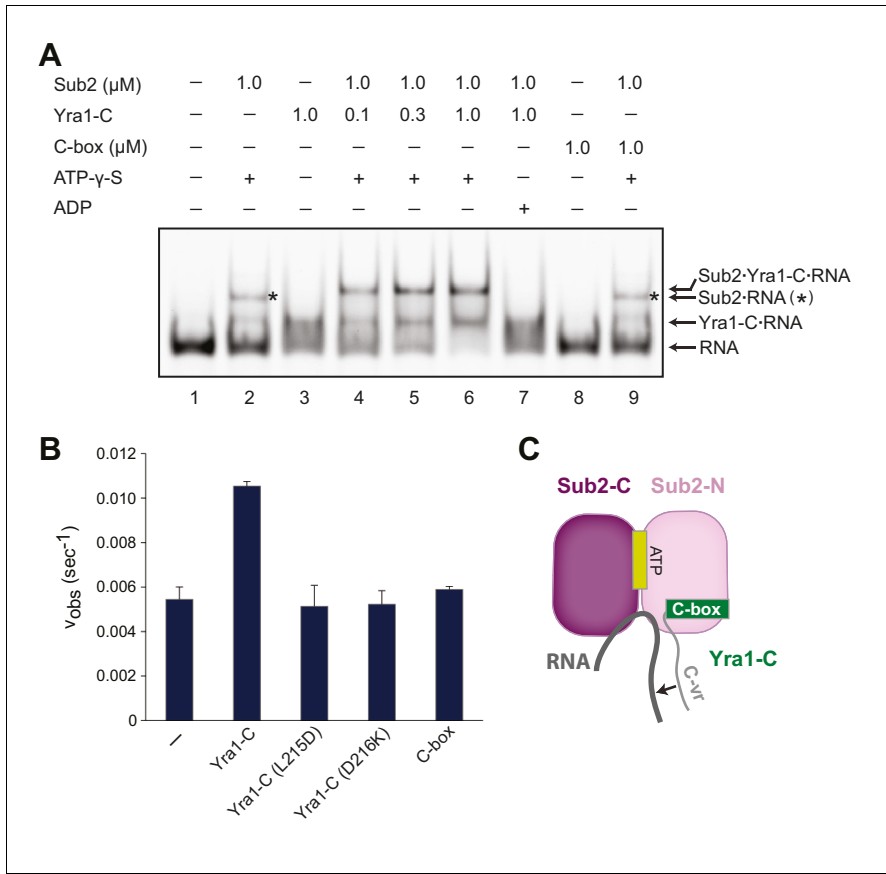

**Figure 4.** Sub2 and Yra1 cooperatively bind to RNA. (**A**) Effect of Yra1 on the RNA binding properties of Sub2. An electrophoretic mobility shift assay was carried out with a fluorescently labeled poly(U) 20-mer RNA. Note that the Sub2•RNA complex (marked with a star) migrated slightly faster than the Sub2•Yra1-C•RNA complex. (**B**) Disruption of either Yra1-Sub2 or Yra1-RNA interaction compromises the stimulation of Sub2 ATPase activity by Yra1. Error bars represent standard deviation of three independent experiments. The reaction rate for Sub2 alone is the same as that in *Figure 2B*. (**C**) Schematic representation of the interaction between Sub2, Yra1-C, and RNA. The arrow indicates a potential interaction between the C-vr region of Yra1 and RNA, based on biochemical studies in **A** and the position of C-box in the Sub2•Yra1-C*•RNA structure.

(*Figure 4A*, compare lanes 2 and 4–6). Importantly, this stimulation of RNA binding to Sub2 by Yra1-C was dependent on the presence of both ATP-γ-S and the C-vr region of Yra1 (*Figure 4A*, compare lanes 6, 7, and 9). Our structural and biochemical studies suggest that Sub2 and Yra1-C, by spatially juxtaposing their RNA binding regions, cooperatively bind to RNA.

## Yra1 stimulates the ATPase activity of Sub2

Previous reports showed that human Alyref (Yra1 in yeast) is an activator of the UAP56 (Sub2 in yeast) ATPase (*Chang et al., 2013*; *Taniguchi and Ohno, 2008*). To test whether this mechanism is conserved in yeast, we examined the effect of Yra1-C on the ATPase activity of Sub2. We found that Yra1-C is capable of accelerating Sub2 ATPase activity by ~2 fold (*Figure 4B*), which is comparable to the reported ~3 fold UAP56 stimulation by full length Alyref. Since Yra1-C interacts with both Sub2 and RNA within the Sub2•Yra1-C•RNA complex, we next assessed the contribution of each interaction to Sub2 stimulation. L215D or D216K mutation was introduced on Yra1 to, respectively, disrupt the hydrophobic or polar interactions at the Sub2-Yra1 interface (*Figure 3D*). We found that both mutations abolished the stimulatory effect of Yra1-C (*Figure 4B*). Similarly, C-box alone, lacking the C-vr RNA binding region, showed no stimulation of Sub2. Taken together, we conclude that Yra1-C exhibits a bipartite interaction, with the C-box binding to Sub2-N and the C-vr binding to RNA, both of which contribute to Sub2 activation (*Figure 4C*).

To further examine how THO and Yra1 act together to modulate the activity of Sub2, we performed the ATPase assay in the presence of both THO and Yra1-C (*Figure 2—figure supplement 2*). Interestingly, the Sub2 ATPase activity was enhanced by ~4.2 fold, a stronger effect than THO or Yra1-C alone. Our results suggest that THO and Yra1 act cumulatively to regulate Sub2.

It should be noted that RNA binding of Sub2 is transient during the ATPase cycle, and could only be detected when turnover of ATP is inhibited. The ADP•BeF$_3$-trapped stable trimeric complex of Sub2, Yra1-C, and RNA, represents a key snapshot for the ATP-dependent recruitment of Yra1 by Sub2 that occurs on the mRNP. At this stage, Yra1 is engaged with the transcript via the C-vr region. ATP hydrolysis would then trigger release of Sub2, whose position on the mRNP might be taken by other RNA binding regions (N-vr and RRM) embedded in Yra1. Our studies on the Sub2-Yra1 interaction clearly indicate that the Sub2 ATP hydrolysis cycle is coupled to loading of Yra1 onto the mRNP.

Some other proteins have been shown to contain C-box motifs, such as Uif, Chtop, and Luzp4, all of which are capable of binding RNA (*Chang et al., 2013*; *Hautbergue et al., 2009*; *Viphakone et al., 2015*). Among them, Chtop has been reported to stimulate UAP56 ATPase activity (*Chang et al., 2013*). UAP56 could use the same mechanism to load these factors, in addition to Alyref, onto mRNPs. Like Sub2 and Yra1, a similar mode of interaction has been reported for the translation initiation factor eIF4A and its activators eIF4B and eIF4H (*Rozovsky et al., 2008*). eIF4B and eIF4H bind to the same region of eIF4A, which includes the analogous region of Sub2 recognized by Yra1. Furthermore, eIF4B and eIF4H make additional transient interactions with RNA, and so contribute to RNA affinity of their complexes with eIF4A. The mechanism by which eIF4B and eIF4H stimulate eIF4A could resemble that of Sub2 stimulation by Yra1. Thus, C-box containing proteins, together with eIF4B and eIF4H, likely represent a new class of DEAD-box ATPase regulators in addition to the MIF4G domain-containing activators.

## Discussion

The TREX complex plays a critical role in the recruitment of export factors to produce export competent mRNPs (*Rodríguez-Navarro and Hurt, 2011*; *Stewart, 2010*). Here, our structural observations combined with biochemical studies have provided molecular snapshots of these TREX-mediated mRNP remodeling reactions (*Figure 5*). Our data support a model in which THO mediates targeted deposition of Yra1 and Mex67•Mtr2 on mRNPs through an ATP-dependent process.

Previous studies suggested that the Sub2 ATPase is recruited to the transcription machinery via the THO complex, which directly binds phosphorylated Pol II CTD (*Meinel et al., 2013*; *Zenklusen et al., 2002*). In doing so, THO would bring Sub2 into spatial proximity to an elongating mRNA transcript. Our studies indicate that THO binding serves as more than a physical attachment. In the complex with THO, Sub2 assumes a semi-open conformation, which is similar to its catalytically active state. Interestingly, we found that THO stimulates the ATPase activity of Sub2. Our data

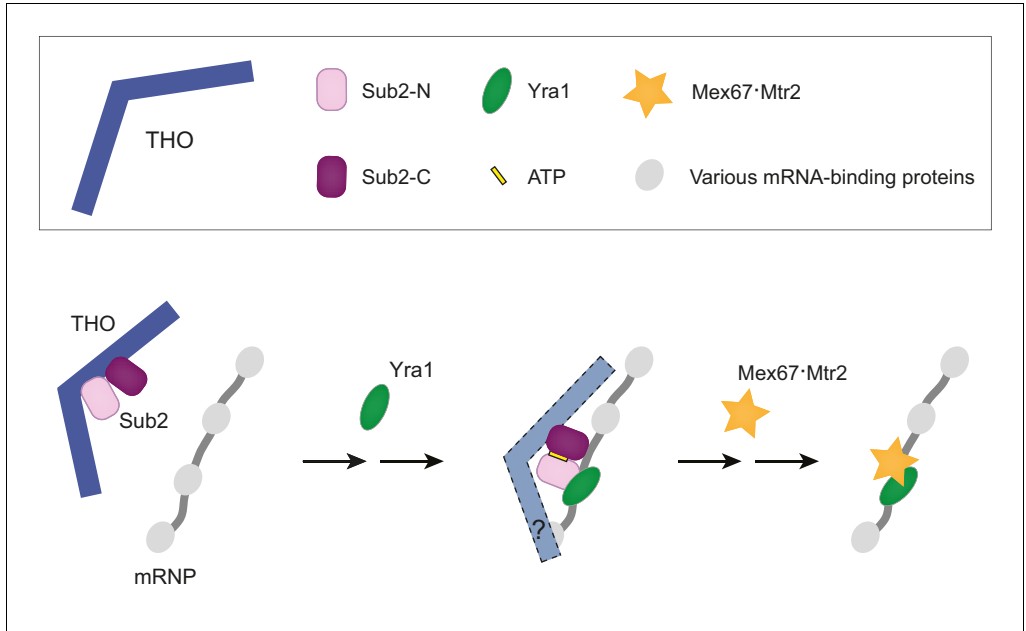

**Figure 5.** A working model of TREX-mediated mRNP remodeling prior to nuclear export. THO recognizes some landmarks of properly processed mRNP, primes Sub2 in a semi-open configuration and mediates targeted deposition of Sub2 and Yra1 onto the mRNP. Next, THO and Yra1 assist in loading of the export receptor Mex67•Mtr2, thereby initiating a series of obligatory reactions for mRNP nuclear export.

support a view that THO, by configuring a semi-open conformation of Sub2, primes it for engaging with mRNA. The mechanism that restricts the activity of TREX to properly processed transcripts is currently not clear. Potentially, THO could deposit Sub2 upon recognizing a certain feature of an mRNP. Previous studies have shown that TREX functions genome-wide, for both intronless and intron-containing genes (*Abruzzi et al., 2004*; *Lei et al., 2011*; *Masuda et al., 2005*). However, it remains to be determined whether THO recognizes a universal feature for both types of transcripts.

The nature of the specific remodeling events on nuclear mRNP triggered by Sub2 is not yet understood. Our biochemical and structural analyses revealed the presence of a stable Sub2•Yra1-C•RNA complex, suggesting that the ATP-dependent binding of Sub2 to an mRNP is coupled to Yra1 loading. Since THO and Yra1 cumulatively stimulated Sub2 ATPase activity in vitro (*Figure 2—figure supplement 2C*), THO and Yra1 could act together to load Sub2 and Yra1 onto the mRNP. THO likely stays on site as it assists in loading of Mex67•Mtr2 in subsequent remodeling reactions (see discussion below). Previous studies indicated that TREX function is connected to pre-mRNA 3'-end processing through the interaction between Yra1 and Pcf11, the Pol II CTD binding subunit of cleavage-polyadenylation factor CF1A (*Johnson et al., 2009*, *2011*). It has been suggested that Yra1 is recruited to the transcription machinery by Pcf11, and then undergoes a concerted deposition with Sub2 onto the mRNP. The precise mechanism of this cross-talk between pre-mRNA 3'-end processing and TREX-mediated mRNP assembly awaits further studies. It should also be noted that the C-box of Yra1, which mediates Sub2 binding, has been found in other mRNA binding proteins, including Uif, Chtop, and Luzp4, all of which interact with the UAP56 ATPase, RNA, and the export receptor TAP•p15 (Mex67•Mtr2 in yeast) (*Chang et al., 2013*; *Hautbergue et al., 2009*; *Viphakone et al., 2015*). Hence, ATP-dependent deposition of export factors on mRNPs is likely a conserved mechanism for C-box containing proteins in higher eukaryotes.

Because binding of Mex67•Mtr2 and Sub2 to Yra1 is mutually exclusive, recruitment of Mex67•Mtr2 is thought to drive the dissociation of Sub2 from the mRNP (*Strässer and Hurt, 2001*). In addition, THO is also implicated in Mex67•Mtr2 loading via the Hpr1 subunit (*Gwizdek et al., 2006*). Consistently, studies on the human TREX complex showed that both Alyref and Thoc5 (a subunit of human THO) make contacts with TAP•p15, and promote RNA binding to TAP•p15 (*Viphakone et al., 2012*). Therefore, TREX functions as a recruitment platform to bring export

factors in spatial proximity to nascent transcripts, and coordinates multiple steps of mRNP assembly. Our current work provides a starting point for future investigations of TREX-mediated mRNP remodeling prior to nuclear export.

## Materials and methods

### Protein production

Yeast THO complex was expressed in High-Five insect cells by co-infection of recombinant baculoviruses encoding THO subunits as indicated (*S. cerevisiae* proteins, unless otherwise stated). To prepare the full length THO complex, each subunit was tagged with an N-terminal TEV-cleavable His tag, with the exception of Thp2, which was untagged. For crystallization purposes, various chimeric THO complexes containing *S. cerevisiae* and *S. bayanus* THO subunits were examined; a stable chimeric THO core complex (THO*) was successfully crystallized. To prepare THO*, Tho2 (residues 1–1207) and Tex1 (*S. bayanus*, residues 1–380) subunits were expressed with N-terminal TEV-cleavable His tags, while Hpr1 (residues 1–603), Mft1 (residues 1–256) and Thp2 (residues 1–261) were untagged. For negative stain EM, THO* subunits were also N- or C- terminally tagged with maltose-binding protein (MBP).

High-Five cells were harvested 48 hr after infection. Protein purification was carried out at 4°C. The cells were pelleted, resuspended, and sonicated in the lysis buffer containing 50 mM Tris-HCl (pH 8.0), 300 mM NaCl, 10 mM imidazole, 1 mM phenylmethylsulfonyl fluoride (PMSF), 5 mg/L aprotinin, 1 mg/L pepstatin, 1 mg/L leupeptin, and 0.5 mM Tris(2-carboxyethyl)phosphine (TCEP). THO complexes used for crystallization and biochemical studies were purified by $Ni^{2+}$-affinity chromatography, followed by TEV digestion to remove N-terminal His tags. MBP-tagged THO complexes for EM studies were purified on amylose resin. Following affinity chromatography, the protein complexes were further purified on a mono Q anion-exchange column (GE Healthcare, Piscataway, NJ) and subjected to size exclusion chromatography using a Superose 6 column (GE Healthcare) in 10 mM Tris-HCl (pH 8.0), 150 mM NaCl, and 0.5 mM TCEP. Purified proteins were concentrated, flash frozen in liquid nitrogen, and stored at −80°C.

Sub2 and Yra1 variants (other than Yra1 C-box) were expressed in *E. coli* BL21-CodonPlus(DE3)-RIL cells (Stratagene), with an N-terminal TEV-cleavable GST tag. Site-directed mutagenesis was done by overlap extension PCR. Protein expression was induced at an $OD_{600}$ of 1.0 with 0.5 mM isopropyl-$\beta$-D-thiogalactoside (IPTG) at 20°C for 16 hr. Cells were pelleted, resuspended, and sonicated in lysis buffer. GST-tagged proteins were purified using a glutathione sepharose four fast flow affinity column (GE Healthcare). After removing the GST tag by TEV, samples were applied to a mono Q column (for Sub2 variants) or a mono S column (GE Healthcare, for Yra1 variants), followed by size-exclusion chromatography using a Superdex 200 column (GE Healthcare) in 10 mM Tris-HCl (pH 8.0), 150 mM NaCl, and 0.5 mM TCEP. Purified proteins were concentrated, flash frozen in liquid nitrogen, and stored at −80°C. C-box (a.a. 208–226) of Yra1 was synthesized by GenScript.

### Crystallization and data collection of the THO*•Sub2 complex

THO* complex and Sub2 were mixed at a molar ratio of 1:1.5 in the presence of 2 mM $MgCl_2$ and 1 mM ADP. Crystals of the THO*•Sub2 complex were obtained at 16°C by sitting-drop vapor diffusion using 1 µl of the protein mixture, 1 µl of reservoir solution (100 mM Tris-HCl, pH 8.4, 17.5% PEG3350, 2% methanol), and 0.2 µl of 3.0 M NDSB-195 (Hampton Research, Aliso Viejo, CA). Streak seeding was required to produce good-quality crystals.

To prepare phosphotungstate cluster ($Na_3[PW_{12}O_{40}]$, Jena Bioscience, Germany) derivative crystals, the solution in the crystallization drop was exchanged into 100 mM HEPES (pH 7.5), 17.5% PEG3350; $Na_3[PW_{12}O_{40}]$ dissolved in this buffer was added to the drop at a final concentration of 0.5 mM. After 3 days, protein crystals were transferred in three steps of increasing PEG400 concentration to cryoprotectant solution (100 mM HEPES, pH 7.5, 17.5% PEG3350, 20% PEG400) and were flash frozen in liquid nitrogen. To prepare tantalum bromide cluster ($[Ta_6Br_{12}]Br_2$, Jena Bioscience) derivatives, crystals were soaked with 1 mM $[Ta_6Br_{12}]Br_2$ in 100 mM Tris-HCl (pH 8.4), 17.5% PEG3350. After 1 day, protein crystals were transferred in three steps of increasing PEG400 concentration to the cryoprotectant solution (100 mM Tris-HCl, pH 8.4, 17.5% PEG3350, 20% PEG400) and were flash frozen in liquid nitrogen.

X-ray diffraction data were collected at beamline ID24-C at APS, as well as beamline X25 and X29 at NSLS. All data were processed using the HKL suite (*Table 1*) (*Otwinowski and Minor, 1997*). Crystals of the THO*•Sub2 complex belong to the P2₁ space group with unit-cell dimensions ~170 × 340 × 175 Å. The crystals typically diffracted to 6–8 Å resolution. Heavy atom soaking, despite shrinking the unit-cell to ~150 × 320 × 175 Å, did not improve the resolution.

## Structure determination of the THO*•Sub2 complex

Initial phases were obtained by single-wavelength anomalous diffraction (SAD) using Phenix (*Adams et al., 2010*). Experimental maps were calculated independently from three crystals (W-I, W-II and W-III) soaked with phosphotungstate cluster and two crystals (Ta-I and Ta-II) soaked with tantalum bromide cluster. The heavy atom substructures enabled us to identify a two fold non-crystallographic symmetry (NCS) of the THO*•Sub2 crystals by PROFESSS (*Collaborative Computational Project, Number 4, 1994*). The resulting electron density maps were of high enough quality that a seven-bladed β-propeller, corresponding to the Tex1 subunit, was readily recognizable. Consistent with the secondary structure prediction of the other THO subunits, rod like electron densities, corresponding to α-helices, constitute the majority regions of the electron density maps.

We used the experimental map calculated from W-II to carry out the first round of model building. Ideal polyalanine α-helices were fitted into rod like densities, where applicable, with COOT (*Emsley and Cowtan, 2004*). Sub2 N-terminal and C-terminal RecA-like domains, from the Sub2 structure reported in this study (*Figure 3B*), were independently positioned into the experimental density map by Molrep (*Vagin and Teplyakov, 1997*). The Tex1 subunit of THO* was modeled by the β-propeller protein WDR5 (PDB 2CNX), which shares 41% sequence homology with *S. bayanus* Tex1 for the residue range 50–344. WDR5, with all the loops connecting β−strands deleted and the remaining residues changed to alanines, was positioned into the experimental density map by Molrep (*Vagin and Teplyakov, 1997*). The seven fold rotational symmetry of the Tex1 propeller prevents us from a definitive assignment of its N/C termini, therefore, the Tex1 model here is meant to show the overall folding.

The initial model allowed us to generate domain masks for subsequent multi-crystal, multi-domain NCS averaging and phase extension using DMMULTI (*Cowtan, 1994*). The THO*•Sub2 model was separated into multiple domains by inspecting the difference in experimental maps from different heavy atom derivatives. The five datasets mentioned above (W-I, W-II, W-III, Ta-I, and Ta-II) were employed for density modification. The resulting map, with an extended phase to 6.0 Å, significantly reduced the noise and improved continuity of electron densities. The THO*•Sub2 model was rebuilt accordingly with COOT (*Emsley and Cowtan, 2004*).

We carried out rigid body refinement at 6.0 Å using Phenix (*Adams et al., 2010*). Eleven rigid bodies were defined, including Sub2 N-terminal domain, Sub2 C-terminal domain, Tex1, as well as eight THO domains. The map after refinement was highly similar to that from the experimental phases. Helices were trimmed/extended and/or centered to fit the *2Fo-Fc* map. The Sub2 N-terminal extension (residues 1–61) is likely disordered because no electron density was observed to connect to the existing N-terminus of the Sub2 model. It is not clear whether ADP is bound to Sub2, although it was required for obtaining the THO*•Sub2 crystals. At this resolution, many of the loops connecting the helices are not visible, therefore the subunit identity is not defined other than for Sub2 and Tex1. To validate the model, we generated an omit map which showed electron density at the expected positions of omitted helices (*Figure 1—figure supplement 1*). Molecular graphics were generated using PyMOL (http://www.pymol.org).

## Crystallization and structure determination of the Sub2•Yra1-C*•RNA complex

Sub2 (a.a. 62–446), Yra1-C* (a.a. 200–226) and poly (U) 15-mer RNA were mixed at a molar ratio of 1:1.2:1.2 in the presence of 2 mM MgCl₂ and ADP•BeF₃ (prepared in a 1:4:20 ratio of ADP:Be:F). Crystals were grown at 4°C by sitting-drop vapor diffusion using a reservoir solution consisting of 100 mM Bis-Tris (pH 5.5), 20% PEG3350, and 0.2 M ammonium acetate. Crystals were transferred in three steps of increasing PEG400 concentration to cryoprotectant solution (100 mM Bis-Tris, pH 5.5, 20% PEG3350, 0.2 M ammonium acetate, 20% PEG400) and were flash frozen in liquid nitrogen.

X-ray diffraction data were collected at beamline ID24-C at APS. Data were processed using XDS and AIMLESS (*Table 2*) (*Collaborative Computational Project, Number 4, 1994*; *Kabsch, 2010*). The structure was solved using Phenix AutoMR by searching for three copies of each RecA-like domain of UAP56 (PDB 1XTJ, human Sub2) (*Adams et al., 2010*). Model was rebuilt in COOT from this molecular replacement solution, followed by refinement with Phenix at 2.6 Å resolution (*Adams et al., 2010*; *Emsley and Cowtan, 2004*). The final Sub2•Yra1-C*•RNA structure contains Sub2 (a.a. 62–444), Yra1 (a.a. 208–224), and poly(U) 6-mer RNA. Molecular graphics were generated using PyMOL (http://www.pymol.org).

## Electron microscopy and image processing

Samples were prepared by negative staining with 0.75% (w/v) uranyl formate (*Ohi et al., 2004*). Data were collected with a Tecnai F20 (FEI) equipped with a Schottky field emission gun and operated at an acceleration voltage of 120 kV. SerialEM (*Mastronarde, 2005*) was used for automatic data collection using low-dose procedures at a calibrated magnification of 88,000× and a defocus value of −2.0 µm. Images were recorded on a 4K × 4K CCD Tietz camera with a binning factor of 2, corresponding to a pixel size of 3.4 Å on the specimen level.

A total of 8070 THO*, 6988 MBP-Tho2 THO*, 6501 Tho2-MBP THO*, 5288 MBP-Hpr1 THO*, 8321 Hpr1-MBP THO*, 7492 MBP-Mft1 THO* and 8394 MBP-Thp2 THO* particles were manually selected from the raw micrographs using BOXER (*Ludtke et al., 1999*). Following contrast transfer function correction with the EMAN2 software package (*Tang et al., 2007*), these particles were rotationally and translationally aligned and subjected to reference-free alignment and K-means classification by SPIDER (*Frank et al., 1996*) specifying 100–125 output classes.

## ATPase assay

Steady-state ATPase activity was analyzed using an NADH enzyme-coupled absorbance assay (*Montpetit et al., 2012*). Briefly, standard ATPase reactions were prepared with indicated proteins (2 µM for Sub2 or THO, 10 µM for Yra1), in 20 mM HEPES (pH 7.0), 100 mM NaCl, 2 mM $MgCl_2$, 1 mM TCEP, 10 µM poly(U) 20-mer RNA (unless otherwise stated), 1 mM ATP, 2 mM phosphoenolpyruvate, 0.2 mM NADH, and 1% (vol/vol) pyruvate kinase/lactate dehydrogenase (Sigma). UV absorbance at 340 nm was monitored by a BioTek Synergy NEO Microplate Reader at 37°C. Reaction rates were calculated from the slopes of the linear phase showing the decrease in NADH absorbance as a function of time.

## Electrophoretic mobility shift assay

For the experiment in *Figures 4A* and 20 nM poly(U) 20-mer RNA labeled with Alexa488 at the 5' end was mixed with Sub2, Yra1 variants, and 1 mM nucleotide as indicated in a buffer containing 20 mM HEPES (pH 7.0), 100 mM NaCl, 1 mM TCEP, 2 mM $MgCl_2$, 5% glycerol, and 0.5 U/µl SUPERase-•In RNase Inhibitor (Thermo Fisher Scientific, Waltham, MA). The mixtures were incubated at room temperature for 10 min. Samples were separated on a 5% native PAGE gel that was prepared with 45 mM Tris-borate. RNA was visualized with the Typhoon 9400 Variable Mode Imager (GE Healthcare).

## Accession numbers

The atomic coordinates and structure factors for the reported crystal structures are deposited in the Protein Data Bank under accession codes 5SUP and 5SUQ.

## Acknowledgements

We thank Annie Heroux at the National Synchrotron Light Source beamline X25 and X29, as well as Raj Rajashankar and Sukumar Narayanasami at Advanced Photon Source beamline 24ID-C for assistance with X-ray data collection; Edward Eng and William Rice at the New York Structural Biology Center (NYSBC) for help with EM data collection and analysis; Phil Jeffrey for support with the X-ray structure determination; Elias Coutavas and Xiaochun Li for critical comments on the manuscript. The data collected at NYSBC was made possible by a grant from NYSTAR. The investigation was conducted in a facility constructed with support from Research Facilities Improvement Program

Grant number C06RR017528-01-CEM from the National Center for Research Resources, National Institutes of Health. YR is supported by the Women and Science Fellowship of the Rockefeller University.

## Additional information

### Funding

| Funder | Author |
| --- | --- |
| Howard Hughes Medical Institute | Yi Ren<br>Philip Schmiege<br>Gunter Blobel |

The funders had no role in study design, data collection and interpretation, or the decision to submit the work for publication.

### Author contributions

YR, Conceptualization, Data curation, Formal analysis, Writing—original draft, Writing—review and editing; PS, Data curation, Formal analysis, Writing—review and editing; GB, Conceptualization, Formal analysis, Funding acquisition, Writing—original draft, Writing—review and editing

### Author ORCIDs

Günter Blobel, http://orcid.org/0000-0002-7839-8341

## Additional files

### Major datasets

The following datasets were generated:

| Author(s) | Year | Dataset title | Dataset URL | Database, license, and accessibility information |
| --- | --- | --- | --- | --- |
| Ren Y, Schmiege P, Blobel G | 2016 | Structure of mRNA export factors | http://www.rcsb.org/pdb/explore/explore.do?structureId=5SUP | Publicly available at the RCSB Protein Data Bank (accession no. 5SUP) |
| Ren Y, Blobel G | 2016 | Structure of mRNA export factors | http://www.rcsb.org/pdb/explore/explore.do?structureId=5SUQ | Publicly available at the RCSB Protein Data Bank (accession no. 5SUQ) |

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
