## [Decision Letter]

Thank you for submitting your article "Structural remodeling of mRNP prior to nuclear export" for consideration by *eLife*. Your article has been favorably evaluated by James Manley as the Senior Editor and three reviewers, one of whom is a member of our Board of Reviewing Editors. The reviewers have opted to remain anonymous.

The reviewers have discussed the reviews with one another and the Reviewing Editor has drafted this decision to help you prepare a revised submission.

Summary:

In this manuscript, Ren and colleagues describe a low-resolution crystal structure of the THO complex bound to the DEAD-box ATPase Sub2, and a structure of Sub2 bound to a small peptide derived from the very C-terminus of Yra1. In addition, the authors perform ATPase and RNA binding assays to show that both the THO complex and the C-terminus of Yra1 stimulate the ATPase activity of Sub2, and that the C-terminal Yra1 peptide enhances RNA binding of Yra1.

While the resolution obtained for the THO complex is low, this manuscript represents a considerable advance and will serve as a starting point for a broad range of experiments aimed at establishing the precise molecular mechanism by which export-competent mRNPs are generated. Therefore, this manuscript is in principle suitable for *eLife*. However, revisions will be necessary before the manuscript can be accepted for publication.

Essential revisions:

1) The resolution of the THO-Tex1-Sub2 is low and the R_free_ is very high (43%) and this structure could therefore be considered preliminary. It would be desirable if this could be improved.

2) The crystal structure and negative stain EM map show different orientation of Tex1. It looks as if the domain is pointing in the opposite direction. The authors should include a low pass filtered map of the reported crystal structure for better comparison with negative stain reconstruction. As of now, it is difficult to compare them and judge the similarities.

3) The authors should provide a MolProbity score and comment on the geometry of the RNA.

4) Several experiments lack controls (specified in A-C below), and these should be included to estimate the background noise vs the signal the authors are interpreting. The quality of all the proteins used in the biochemical assays should also be shown in a Coomassie-stained gel to estimate their purity (a potential problem particularly in the case of the unstructured proteins used in the assay).

A) Experiment in Figure 2: should include the controls with THO alone and also Sub2 without RNA to estimate the background ATP hydrolysis. The corresponding curves (raw data) should be shown, either in the main text or in supplementary material.

B) Experiment in Figure 4: should include controls of Yra1 constructs without Sub2 (to estimate the background activity in the preps). Is the background noise significantly lower than the signal? Is it possible that the proteins co-purify with a chaperone or other impurity, and that the impurity contributes to the activity? Again, the original curves should be shown.

C) Experiment in Figure 2—figure supplement 2: should include controls with THO alone and Yra1-C alone. The original curves should be shown. From this experiment, the authors state that THO and Yra1-C synergistically accelerate the ATP hydrolysis rate of Sub2. However, the data suggest that the effect is additive, not synergistic.

5) It is unclear how the two parts of the paper fit together and how Sub2 can bind both activators simultaneously. This is an obvious aspect that the authors should be able to analyze, even with their current data. Also, it is unclear how the results in the paper are interpreted in the final model. In the model, ATP is bound to Sub2-THO-Yra1 and it is hydrolyzed upon binding of Mex67•Mtr2. However, in the paper the authors show that THO and Yra1 increase the ATPase activity of Sub2. There is no evidence in the manuscript for a role of Mex67•Mtr2 in the ATP-hydrolysis step. In addition, what is the evidence that Sub2 is dissociating from TREX at the very end and not before? If the N-box and C-box both bind and are partially redundant, why can they not bind to both Sub2 and Mex67•Mtr2 at the same time?

6) The title is inflated and should be changed to fit the actual take home message of the paper, namely that Sub2 is activated by THO and Yra1.

---

## [Author Response]

*Essential revisions:*

*1) The resolution of the THO-Tex1-Sub2 is low and the R_free_ is very high (43%) and this structure could therefore be considered preliminary. It would be desirable if this could be improved.*

We feel it’s appropriate to point out the technical challenge to determine a crystal structure of a 360 kDa protein complex with little prior structural information. Many reported crystal structures at low resolutions are determined by molecular replacement and refined with existing atomic structures, while experimental phasing and ab initio modeling at similar resolutions usually generate very high R_free_. For example, the R_free_ of a 6.6 Å poly-alanine model of the human DNA-dependent protein kinase (DNA-PK) is 44% (PDB ID 3KGV); the R_free_ of the dynein motor domain structure at 6.0 Å resolution is 43% (PDB ID 3QMZ). Therefore, the R_free_ of our THO*•Sub2 model is comparable to previously reported structures determined by similar means. We hope to provide a THO•Sub2 structure at a better resolution in our future studies.

*2) The crystal structure and negative stain EM map show different orientation of Tex1. It looks as if the domain is pointing in the opposite direction. The authors should include a low pass filtered map of the reported crystal structure for better comparison with negative stain reconstruction. As of now, it is difficult to compare them and judge the similarities.*

The THO model low-pass filtered at 40 Å using EMAN2 is now shown in Figure 1—figure supplement 2. The reported crystal structure contains THO* and Sub2, while negative stain EM was carried out with THO* alone. The different orientation of Tex1 may result from conformational change of THO* upon Sub2 binding, or may be induced by crystal packing.

The purpose of the EM studies is to locate each subunit of THO* through MBP labeling. The similarity between the THO* model from the crystal structure and 2D class averages of negatively stained THO* allows us to unambiguously determine the THO* subunit arrangement.

*3) The authors should provide a MolProbity score and comment on the geometry of the RNA.*

The protein geometry was analyzed using MolProbity as the reviewers requested. The MolProbity score of the THO*•Sub2 structure is 1.58 (100^th^ percentile, N=342, 3.25 Å – 6.25 Å); and that of the Sub2•Yra1-C*•RNA structure is 1.53 (100^th^ percentile, N=6237, 2.600 Å ± 0.25 Å). It looks like that the MolProbity score of the THO*•Sub2 structure may not be informative in this particular case, because the majority part of the model is built with ideal α helices. Therefore, we feel it is best not to include it.

Regarding the geometry of RNA, we now state the following: “The RNA is sharply bent between U4 and U5, as observed in structures of other RNA bound DEAD-box ATPases.”

*4) Several experiments lack controls (specified in A-C below), and these should be included to estimate the background noise vs the signal the authors are interpreting. The quality of all the proteins used in the biochemical assays should also be shown in a Coomassie-stained gel to estimate their purity (a potential problem particularly in the case of the unstructured proteins used in the assay).*

The control experiments are included in the current manuscript as the reviewers suggested. All the recombinant proteins used in our work have been purified by affinity chromatography, followed by tag removal (except negative stain EM studies), ion-exchange chromatograph and size exclusion chromatography. Regarding the less structured Yra1-C protein, it showed degradation during earlier steps of purification, but fortunately full-length Yra1-C can be purified away from the degradation products using ion-exchange chromatography. The molecular weight of Yra1-C was verified by mass spectrometry. Coomassie-stained gels of all the recombinant proteins are included in Figure 1 and Figure 2—figure supplement 2.

*A) Experiment in Figure 2: should include the controls with THO alone and also Sub2 without RNA to estimate the background ATP hydrolysis. The corresponding curves (raw data) should be shown, either in the main text or in supplementary material.*

Representative raw data of the NADH enzyme-coupled ATPase assay are now included in Figure 2—figure supplement 2. UV reading shows the decrease of NADH absorbance as a result of ADP to ATP conversion. The rate of NADH background decomposition was subtracted from all other reactions to calculate the ATP hydrolysis rate. In our assay, we optimized the concentration of NADH to minimize NADH background signal and to obtain reliable UV reading. Our results indicate that recombinant THO complex did not show ATPase activity, ruling out the possibility of contaminating ATPases (such as chaperones). Sub2 also did not show noticeable ATPase activity in the absence of RNA.

*B) Experiment in Figure 4: should include controls of Yra1 constructs without Sub2 (to estimate the background activity in the preps). Is the background noise significantly lower than the signal? Is it possible that the proteins co-purify with a chaperone or other impurity, and that the impurity contributes to the activity? Again, the original curves should be shown.*

Control experiments are now shown in Figure 2—figure supplement 2. Yra1-C alone did not show ATPase activity. The extent of Sub2 stimulation by Yra1, albeit to a modest extent, is in agreement with previously reported ~ 3 fold stimulation of UAP56 by Alyref.

*C) Experiment in Figure 2—figure supplement 2: should include controls with THO alone and Yra1-C alone. The original curves should be shown. From this experiment, the authors state that THO and Yra1-C synergistically accelerate the ATP hydrolysis rate of Sub2. However, the data suggest that the effect is additive, not synergistic.*

Control experiments are now shown in Figure 2—figure supplement 2. We have changed “synergistically” to “cumulatively” when describing the effect of THO and Yra1 on the ATPase activity of Sub2.

*5) It is unclear how the two parts of the paper fit together and how Sub2 can bind both activators simultaneously. This is an obvious aspect that the authors should be able to analyze, even with their current data. Also, it is unclear how the results in the paper are interpreted in the final model. In the model, ATP is bound to Sub2-THO-Yra1 and it is hydrolyzed upon binding of Mex67•Mtr2. However, in the paper the authors show that THO and Yra1 increase the ATPase activity of Sub2. There is no evidence in the manuscript for a role of Mex67•Mtr2 in the ATP-hydrolysis step. In addition, what is the evidence that Sub2 is dissociating from TREX at the very end and not before? If the N-box and C-box both bind and are partially redundant, why can they not bind to both Sub2 and Mex67•Mtr2 at the same time?*

The model in Figure 5 and the Discussion have been modified. We hope the current manuscript represents our major findings more effectively. Our studies have revealed two states of the Sub2 ATPase during its enzymatic cycle: 1) a half-open state stabilized by THO; and 2) a closed state when bound to RNA and Yra1-C. THO and Yra1-C could in principle bind to Sub2 simultaneously in the presence of RNA. Even if the closed state of Sub2 is not the preferred form bound by THO, THO may latch on to one of the Sub2 domains (presumably with a much lower affinity). We have been unable to obtain a stable complex of THO•Sub2•Yra1-C, but the presence of such a complex cannot be ruled out due to the difficulty in capturing weak/transient interactions. Notably, THO has been shown to interact with the export receptor both in yeast (Mex67•Mtr2) and in humans (TAP•p15), strongly suggesting that THO stays associated with Sub2 until Mex67•Mtr2 is deposited by a yet-as-defined mechanism.

The original figure does not mean that Mex67•Mtr2 plays a role in ATP hydrolysis by Sub2. We apologize for the previous confusing graphical representation. As to whether Yra1 utilizes its N-terminal and C-terminal regions to bind Sub2 and Mex67•Mtr2 simultaneously, there has not been enough evidence to address one way or the other. We hope to address these outstanding questions in future work.

*6) The title is inflated and should be changed to fit the actual take home message of the paper, namely that Sub2 is activated by THO and Yra1.*

Point accepted. We have changed the title to make it more specific: “Structural analysis of helicase-mediated remodeling of nuclear mRNP prior to nuclear export”.